# Robustness of Surface Roughness against Low Number of Picture Elements and Its Benefit for Scaling Analysis

**Wenmeng Zhou [1,2], Xinghui Li [1], Feng Feng [1,2,*], Timing Qu [3], Junlong Huang [1,2], Xiang Qian [1], Huiting Zha [1,2] and Pingfa Feng [1,2,3,*]**

[1] Division of Advanced Manufacturing, Graduate School at Shenzhen, Tsinghua University, Shenzhen 518055, China; zhouwm18@mails.tsinghua.edu.cn (W.Z.); li.xinghui@sz.tsinghua.edu.cn (X.L.); jlhuang13@126.com (J.H.); qian.xiang@sz.tsinghua.edu.cn (X.Q.); zhahuiting123@sz.tsinghua.edu.cn (H.Z.)

[2] Lab of Intelligent Manufacturing and Precision Machining, Tsinghua Shenzhen International Graduate School, Tsinghua University, Shenzhen 518055, China

[3] State Key Laboratory of Tribology, Department of Mechanical Engineering, Tsinghua University, Beijing 100084, China; tmqu@mail.tsinghua.edu.cn

\* Correspondence: feng.feng@sz.tsinghua.edu.cn (F.F.); feng.pingfa@sz.tsinghua.edu.cn (P.F.)

**Abstract:** Surface roughness is widely used in the research of topography, and the scaling characteristics of roughness have been noticed in many fields. To rapidly obtain the relationship between root-mean-squared roughness ($Rq$) and measurement scale ($L$) could be helpful to achieve more understandings of the surface property, particularly the $Rq$-$L$ curve could be fitted to calculate the fractal dimension ($D$). In this study, the robustness of $Rq$ against low number of picture elements was investigated. Artificial surfaces and the surfaces of two actual samples (a silver thin film and a milled workpiece) were selected. When the number of picture elements was lowered, $Rq$ was found to be stable within a large portion of the concerned scope. Such a robustness property could validate the feasibility of $Rq$-$L$ curve obtained by segmenting a single morphological picture with roughness scaling extraction (RSE) method, which was proposed in our previous study. Since the traditional roughness (TR) method to obtain $Rq$-$L$ curves was based on multiple pictures, which used a fixed number of picture elements at various $L$, RSE method could be significantly more rapid than TR method. Moreover, a direct comparison was carried out between RSE method and TR method in calculating the $Rq$-$L$ curve and $D$, and the credibility and accuracy of RSE method with flatten order 1 and 2 was verified.

**Keywords:** surface roughness; robustness; image segmentation; low pixels; fractal dimension

## 1. Introduction

The analysis of surface features is of great significance and is widely applied in relevant fields, such as natural geographical topography [1], and in determining various surface morphology performance [2–4], and evaluating nanostructured-thin film property [5–7]. Zhang et al. [8] and Hosseinabadi et al. [9] reported a series of analyses on the rough surface of multiscale structures by calculating their fractal dimensions. Lawrence et al. [10] reported the fractal characterisation of automotive cylinder linear surfaces, and some researchers [11–13] found that the fractal characteristics of various surface morphologies could affect anisotropy and wear performance. Jing et al. [14] found that the functional properties of Ga-doped ZnO thin films could be evaluated by fractal methods. Talu et al. [15] reported a fractal analysis on the microstructure of ZnO thin films and proposed fractal models to determine the optical and surface roughness of Ag-Cu thin films [16]. With reference to

Mandelbrot's study [17], as the key factor of fractal theory, fractal dimension (*D*) has been widely used to characterise the fragment and self-affine objects, and *D* is independent of scale as stated by Wu et al. [18].

In general, two routes are used to obtain the *D* value of a fractal surface. The first route employs multiple morphological images of the surface, which could be measured through atomic force microscopy (AFM) and white light interferometry (WLI). The measurement scale (*L*) values of the images should be different in a certain range. The root mean square (*Rq*, also known as RMS) surface roughness is calculated, and *D* can be obtained by fitting in accordance with the power-law relationship between *Rq* and *L*, which will be described in detail below. This route was utilised for the analysis of thin films, such as the evaluation of surface roughness of Hastelloy C276 substrates and amorphous alumina buffer layers for high-temperature superconducting coated conductors by Feng et al. [19], surface roughness characteristic of nano-, micro- and ultrafiltration membranes by Wong et al. [20], hydrophobicity studies of nanofiltration membranes under different measured modes by Boussu et al. [21] and self-affine fractal roughness in ion-bombarded film surfaces observed by Krim et al. [22]. The above method will be denoted as the traditional roughness (TR) method in this paper. TR method is based on the scale characteristics of surface roughness, and the accuracy of the roughness value is of significant importance to the result. Therefore, lots of efforts in various research fields have been elaborated to improve the roughness accuracy, such as modelling learning [23] and model-based prediction [24]. Moreover, there are many common standards of surface roughness measurement, such as the definition and calculation standard of *Rq* in ISO 25178 [25], ISO 4287 [26], and the related content of topography roughness measured by AFM in ISO 19606:2017 [27], etc. However, TR method is based on a lot of morphological measurements, thus very long time would be consumed.

The second route is based on a single morphological image for which numerous methods have been proposed and utilised. For example, Ai et al. [28] used the box-counting (BC) method to estimate the *D* of a rock surface, and Li et al. [29] calculated *D* by using the BC method and reported the effect of grinding parameters on the surface features. Ponomareva et al. [30] evaluated *D* of sol-gel deposited oxide films via power spectral density (PSD) method. Sayles et al. [31] found the spatial characteristics of surfaces via the structure function (SF) method, Talu et al. [32] use the autocorrelation function (ACF) to discuss the relationship between topographical surface parameter and monocrystal surface oxidation. However, Kulesza et al. [33] indicated that the accuracy of these single-image based methods is lower than that of the TR method, which is multi-image based.

In our previous study [34], a new single-image based method called roughness scaling extraction (RSE) method was proposed. There were two benefit aspects of RSE method in practical applications. First, less measurement workload was required relative to the multi-image based TR method. Second, the accuracy of RSE method to calculate fractal dimension was found to be better than the traditional single-image based methods. With the use of the artificial fractal surfaces generated through the Weierstrass-Mandelbrot (W-M) function with ideal *D* values, the mean relative error (MRE) between the calculated *D* and the ideal *D* can be calculated, which could quantify the accuracy of a method for *D* calculation. The MRE of the RSE method could be lower than 1%, whilst those of BC, SF and ACF methods ranged in 4–7%. On the basis of its high accuracy, the RSE method was used in our other studies on textured MgO thin films fabricated by energetic particle self-assisted deposition by Feng et al. [35], the influence of morphological filter by AFM probe tip by Feng et al. [36], and fractal trait extraction of electroencephalography signals by Wang et al. [37].

In the RSE method, a single morphological image is segmented into sub-images with small *L*. After flattening modification with a certain order is performed, the *Rq* values of the sub-images are calculated to obtain the *Rq-L* relationship. Typically, the as-measured morphological images have fixed pixels, e.g., 512 × 512 in AFM measurements. Thus, the pixels of a segmented sub-images is lower than that of an image attained by an actual measurement at the same *L*, which may partly decrease the credibility of *Rq* results, as many researchers [38–40] reported in their relevant

studies. Therefore, questions may arise about the similarity and difference between the RSE and TR methods. Particularly, are the *Rq-L* curves obtained by the two methods the same for a fractal surface? Therefore, a reconsideration to validate the RSE method for fractal analysis should be carried out.

In this study, the variation of *Rq* value along with the lowering of the number of picture elements, was investigated. Then, a direct comparison of RSE and TR method was performed. The credibility and application of the RSE method were discussed on the basis of the robustness of *Rq*.

## 2. Materials and Methods

### 2.1. The Analyzed Surfaces

Three types of surfaces were utilised in this study, namely, artificial fractal surfaces, the surfaces of a silver thin film measured by AFM and the surfaces of a milled workpiece measured by WLI. The details of these surfaces are discussed below.

#### 2.1.1. Artificial Fractal Surfaces

Artificial fractal surfaces with an ideal fractal dimension ($D_i$) for analysis were generated by the W-M function, which is self-affine and continuous but non-conductible, as shown as follows in Equation (1), which was reported by Berry et al. [41],

$$
\begin{aligned}
z(x,y) = L\left(\frac{G}{L}\right)^{D_i-2}\left(\frac{\ln\gamma}{M}\right)^{\frac{1}{2}}\sum_{m=1}^{M}\sum_{n=0}^{n_{max}}\gamma^{(D_i-3)n} \\
\left\{\cos(\Phi_{mn}) - \cos\left[\frac{2\pi\gamma^n\left(x^2+y^2\right)^{\frac{1}{2}}}{L}\cos\left(\tan^{-1}(\frac{y}{x}) - \frac{\pi m}{M}\right) + \Phi_{mn}\right]\right\}
\end{aligned}
\tag{1}
$$

$z(x,y)$ is the height of topography at $(x,y)$, $L$ is the measurement scale of surface topography, $G$ is the height scaling coefficient, $\Phi_{mn}$ is a random phase, $D_i$ is ideal fractal dimension which could range from 2 to 3, $\gamma$ is the frequency factor, $n_{max} = \text{int}[\log_\gamma(L/L_s)]$, $L_s$ is the cutoff length, int$[\dots]$ is the largest integer value in square brackets, $n$ ranges from 0 to $n_{max}$, $M$ is the number of overlapping components, $m$ ranges from 1 to $M$. Four fractal surfaces with $D_i$ from 2.2 to 2.8 (interval = 0.2) and whose $L = 80$ µm and surface pixels were 8192 × 8192 were generated and illustrated in Figure 1. When $D_i$ was large, the surface morphology became increasingly irregular and fragmented.

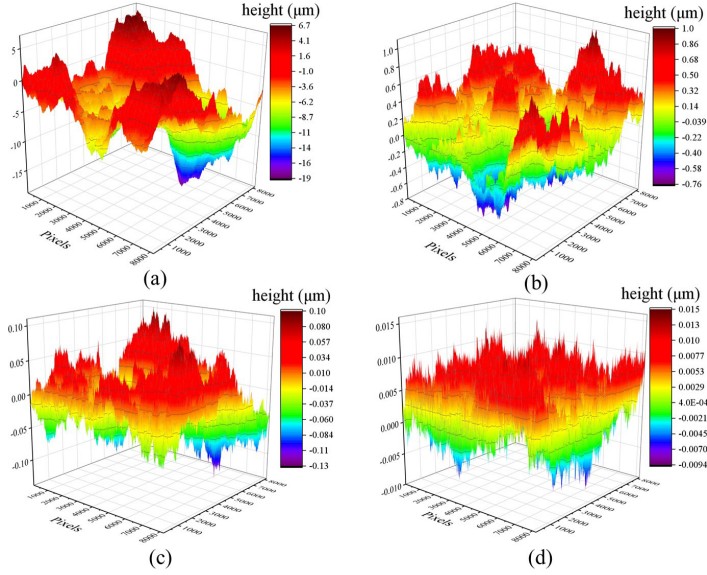

**Figure 1.** Artificial fractal surfaces: (**a**) $D_i$ = 2.2; (**b**) $D_i$ = 2.4; (**c**) $D_i$ = 2.6; (**d**) $D_i$ = 2.8.

### 2.1.2. Surfaces of a Silver Thin Film

As one of the physical vapour deposition methods, magnetron sputtering is commonly utilised in the preparation of thin films [42,43]. Barabasi et al. [44] summarised the fractal characteristics during surface growth. Silver thin films with silicon substrates are prepared using a radio frequency magnetron sputtering system; this process is similar to the fabrication of $CeO_2/YSZ$ buffer layer by Zhang et al. [45] and the deposition of MgO buffer layer by Xiao et al. [46]. During the coating process, the working power is 100 W, the coating time is 12 min, and the substrate temperature is 300 °C. The surface morphologies of the silver thin film measured by AFM (tapping mode) at $L$ of 1 μm are illustrated in Figure 2.

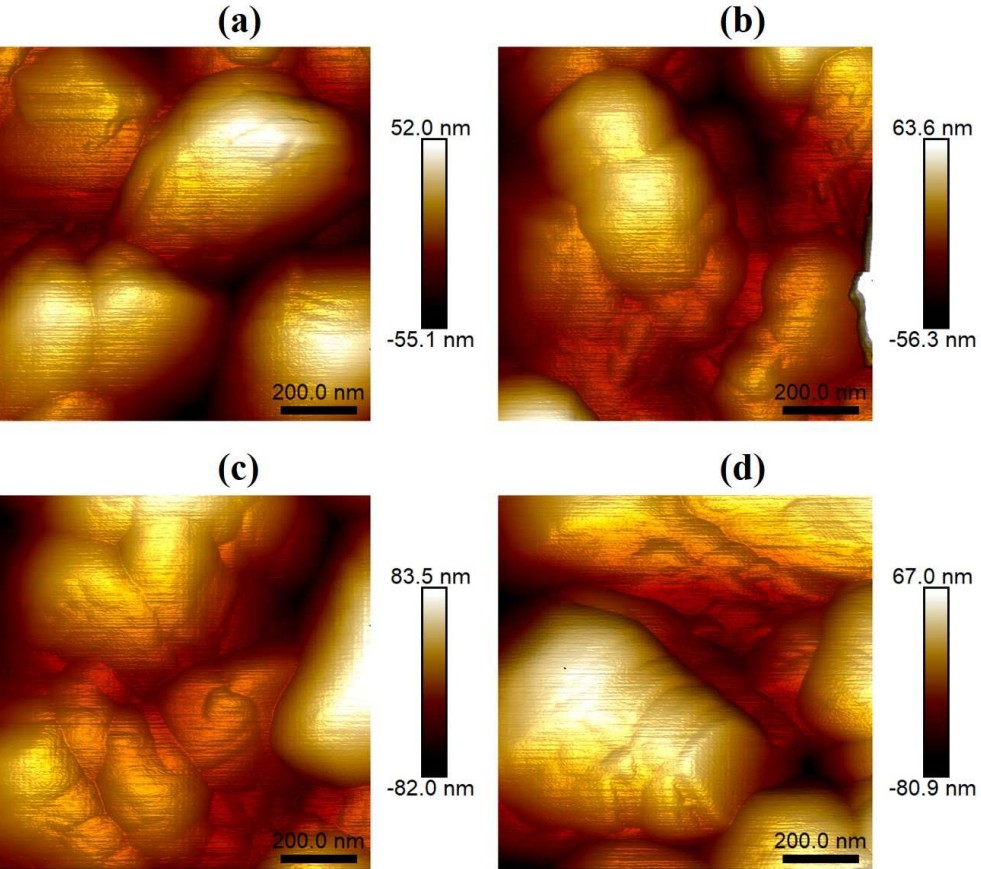

**Figure 2.** AFM images at $L$ of 1 μm of the surfaces of a silver thin film, which was prepared by using magnetron sputtering deposition.The denotations (**a**–**d**) represented four different positions randomly selected on these surfaces, respectively.

### 2.1.3. Surfaces of a Milled Workpiece

Milling is a typical manufacturing technique, and the profiles of machined surface appear random, multiscale and disordered, as reported by Sonbaty et al. [47]. In this study, an aluminum alloy workpiece with a diameter of 6 mm, was milled by a computer numerical control system and a three-edge vertical milling cutter. The spindle speed was 5000 r/min. Four regions of a milled workpiece were randomly collected for analysis, and their morphologies measured by WLI are illustrated in Figure 3.

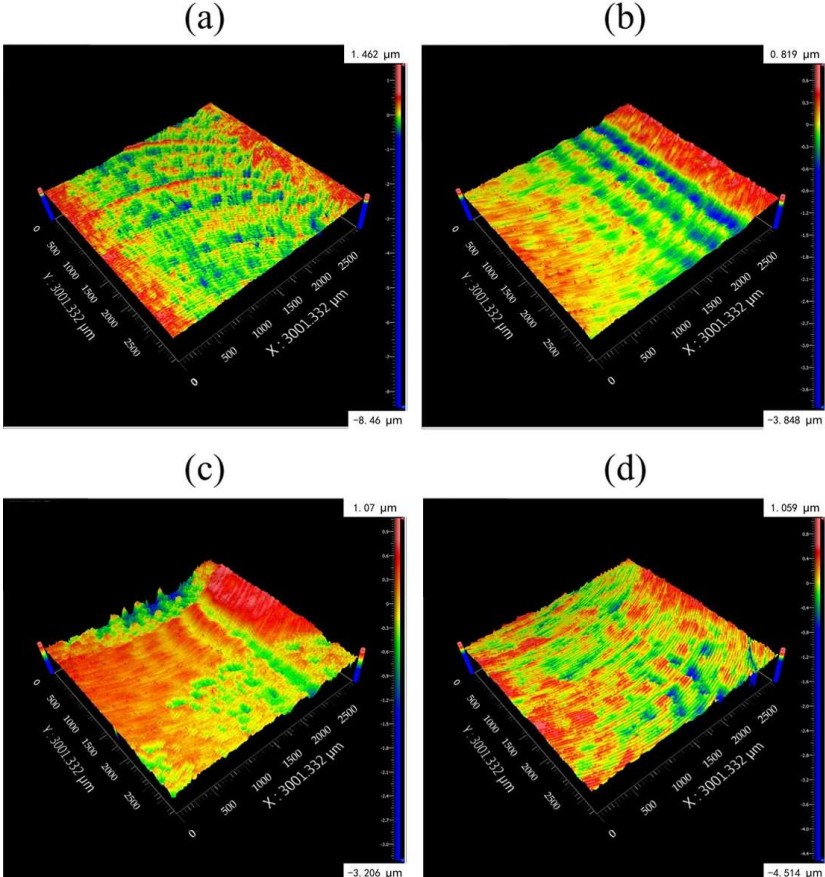

**Figure 3.** WLI images of the surfaces of a milled workpiece. The denotations (**a**–**d**) represented four different positions randomly selected on these surfaces, respectively.

## 2.2. Calculation

### 2.2.1. Sampling of Surface Images

Multiple measurements at the same *L* on the above surfaces were conducted with decreasing number of picture elements to investigate the robustness property of surface roughness against low number of the morphological picture elements (Figure 4).

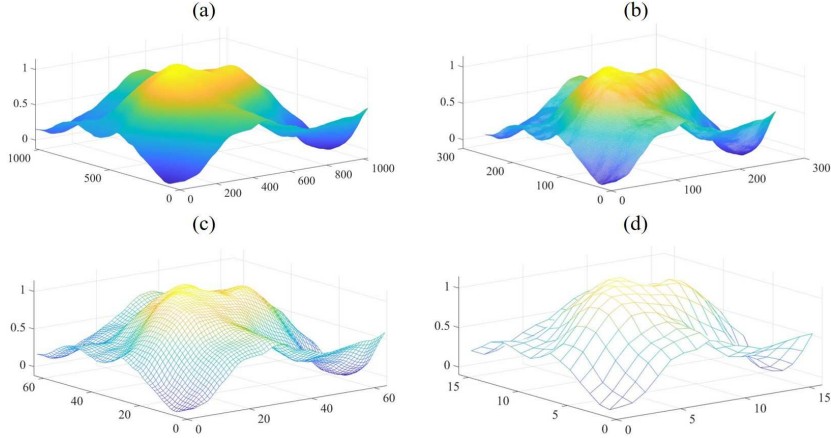

**Figure 4.** Schematic diagram of reducing number of picture elements in sampling process. (**a**) 8192 × 8192; (**b**) 256 × 256; (**c**) 64 × 64; (**d**) 16 × 16.

### 2.2.2. Traditional Roughness (TR) Method

The TR method is a classical method for fractal dimension calculation and is based on the power-law relationship of $Rq$ and $L$, where $H$ is the Hurst exponent, and used in the study by Bigerelle et al. [48], as shown in Equations (2) and (3).

$$Rq = AL^H = AL^{3-D} \tag{2}$$

$$Rq = \left\langle \left( z\left(i,j\right) - \left\langle z\left(i,j\right) \right\rangle \right)^2 \right\rangle^{1/2} \tag{3}$$

In the TR method, the $Rq$ values at various $L$ of a certain surface should be measured to obtain multiple images with fixed pixels, e.g., $512 \times 512$. The $Rq$-$L$ curve, which would be linear in the double logarithm coordinates, is then fitted by using a least-square regression according to Equation (2). Thus, the fractal dimension of the TR method ($D_{TR}$) could be calculated.

Compared with other traditional methods based on a single morphological image (i.e., BC, SF and ACF), the TR method has better accuracy, as reported by Kulesza et al. [33]. However, the TR method is less efficient than other methods because it is based on multiple images.

### 2.2.3. Roughness Scaling Extraction (RSE) Method

RSE method was proposed in our previous study to calculate $D$ accurately. In the operation of RSE method, the sub-images with a certain $L$ were segmented from various parts of the original image, and the $L$ of sub-images were chosen as a geometric series: $q^1L_0, q^2L_0, q^3L_0...$, where $q$ was the segmentation ratio and $L_0$ was the scale of the original image. 100 sub-images were randomly selected at each scale, and then flatten modification was used for these sub-images. The specific process of flatten modification is as follows: The surface topography of a sub-image is extracted line by line into height sequences. Each height sequence is least-squares fitted by using a polynomial with an order of 1 or 2 (i.e., the flatten order), as shown in Equations (4) and (5), respectively. Then the polynomial was subtracted from the height sequence to provide the relative height sequence. Finally, all the relative height sequences are integrated into a new image, which is the flattened sub-image.

$$y = a_1 + b_1 x \tag{4}$$

$$y = a_2 + b_2 x + c_2 x^2 \tag{5}$$

After the flatten modification, $Rq$ value of each sub-image was calculated through Equations (2) and (3), then the average value was calculated as the roughness result at this scale. The calculation processes of the RSE method and the TR method are shown in the following Figure 5. It can be observed that the fundamental difference between the two methods is that the acquisition methods of images of different scales are different.

This method was found to be effective analysing thin film surface and AFM measurement technique by Feng et al. [35,36], and time-series signals such as electroencephalography by Wang et al. [37]. Similar to the TR method, the RSE method is also based on the scaling characteristics of surface roughness. However, the RSE method requires only a single morphological image of the surface.

Instead of the actual measurements of multiple images with fixed pixels at various $L$, sub-images with $L$ smaller than that of the original image were obtained through image segmentations. Thus, the pixels of the sub-images of the RSE method could be lower than that of the image actually measured in the TR method. After segmentation, flattening modification is performed for the sub-images. Then the $Rq$ values were calculated in accordance with Equation (3) to obtain the $Rq$-$L$ curve, and the fractal dimension of RSE method ($D_{RSE}$) could be obtained by fitting the $Rq$-$L$ curve. Miyata et al. [49] did not flatten the sub-images, and the $Rq$-$L$ curve was calculated directly after image segmentation. Such a

case was also considered in this study and was denoted as RSE-f0. The order of flattening modification used in this study includes 1 and 2 and denoted as RSE-f1 and RSE-f2, respectively.

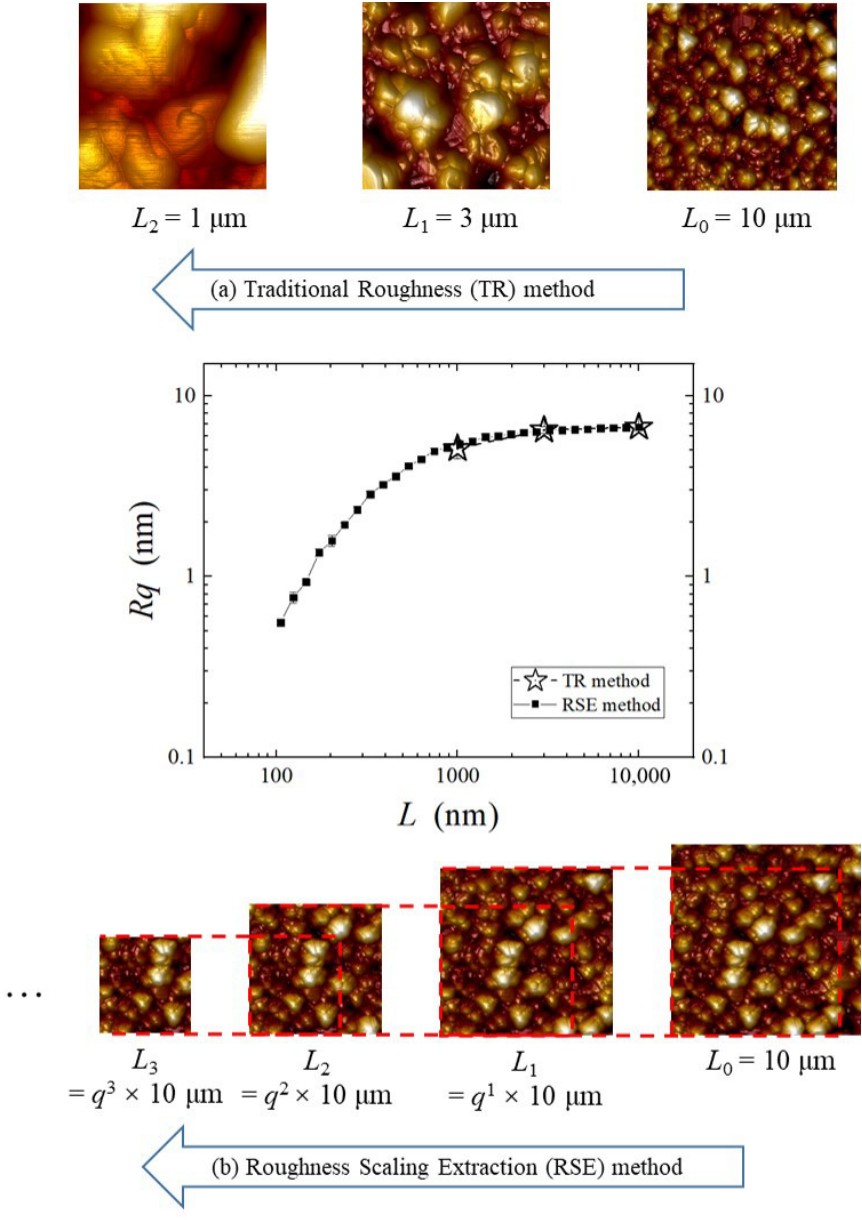

**Figure 5.** Comparison of the operation processes of (**a**) TR method, (**b**) RSE method ($q$ was the segmentation ratio).

## 3. Results

### 3.1. Influence of Low Number of Picture Elements on Rq

The variations of $Rq$ values when the image pixels of the artificial surfaces were altered are illustrated in Figure 6, where the number of one side of the square image is used to qualify the pixels. The $Rq$ values were stable within a large portion of the pixel range for all the curves and generally decreased at low pixels. Therefore, the $Rq$ values at the highest pixels were regarded as the stable roughness value ($Rq_1$) for each case. When the flatten order was increased, the $Rq$ values were lowered, which is consistent with the previous result of our study on thin film surface by Feng et al. [19].

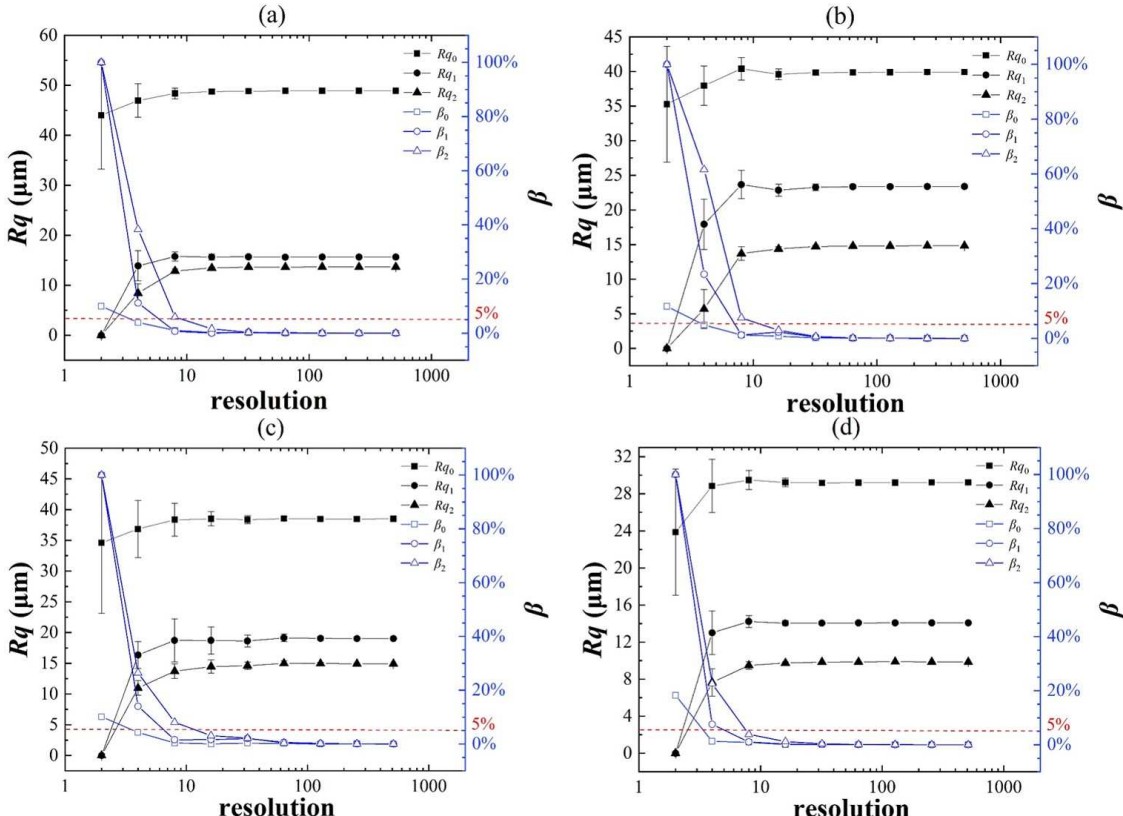

**Figure 6.** *Rq* values obtained by using different number of picture elements and flatten orders for the artificial surfaces shown in Figure 1. The $\beta$ values were calculated through Equation (4) to qualify the *Rq* variations. (**a**) $D_i$ = 2.2; (**b**) $D_i$ = 2.4; (**c**) $D_i$ = 2.6; (**d**) $D_i$ = 2.8.

To qualify the relationship of *Rq* variations and number of picture elements, a parameter $\beta$ was proposed as shown in Equation (4), where $Rq_i$ is the roughness value at $L = i$. The $\beta$ curves are also shown in Figure 6, as indicated by the hollow symbols.

$$\beta = \frac{|Rq_i - Rq_1|}{Rq_1} \times 100\% \tag{6}$$

For all the curves with various $D_i$ values and flatten orders, the $\beta$ value could be close to 0 when the image pixels were above 30. When the flatten order was low, $\beta$ became small at low pixels, indicating that the $Rq_i$ value obtained without flattening modification (i.e., RSE-f0) was the most robust against low number of picture elements.

The variations of *Rq* values for the surfaces of silver thin film and milled workpiece were also analysed. The *Rq* variations obtained using different number of picture elements and the flatten orders of these two surfaces are illustrated in Figures 7 and 8, respectively, and were both similar to the case of artificial surfaces shown in Figure 6.

As shown in Figures 6–8, the criteria of $\beta$ = 5% denoted by the dash line was used as the significant deviation of $Rq_i$ from $Rq_1$. The number of picture elements, where $\beta$ surpassed 5%, was called the minimum sampling number of picture elements, as summarised in Table 1.

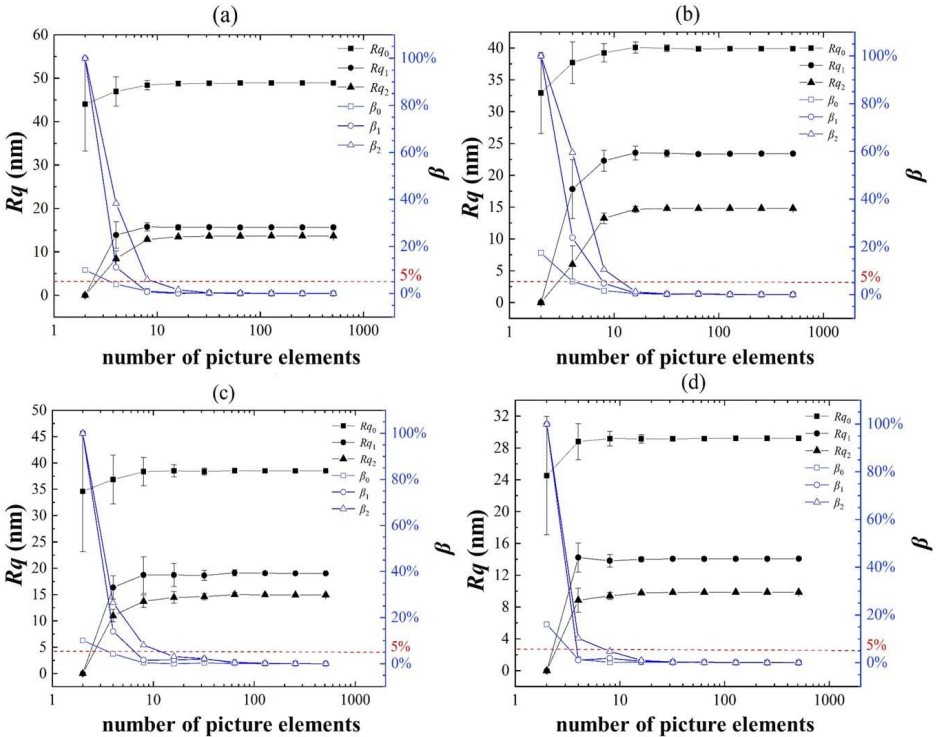

**Figure 7.** *Rq* values obtained by using different number of picture elements and flatten orders for the surfaces of a silver thin film. The subfigures (**a**–**d**) represented four different positions randomly selected on the surface, respectively.

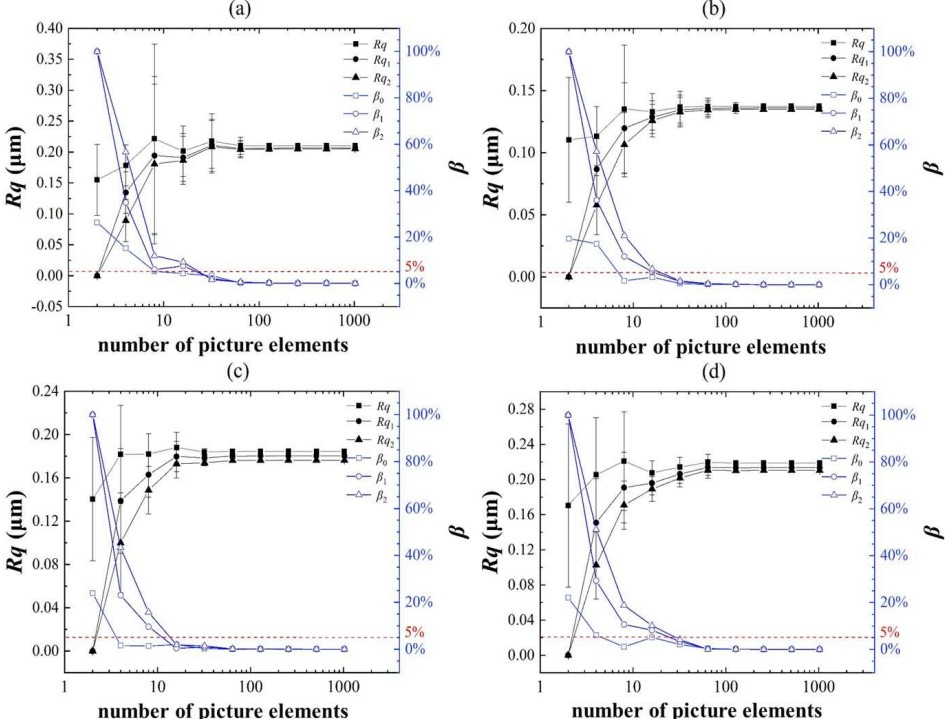

**Figure 8.** *Rq* values obtained by using different number of picture elements and flatten orders for the surfaces of a milled workpiece. The subfigures (**a**–**d**) represented four different positions randomly selected on the surface, respectively.

**Table 1.** The minimum sampling number of picture elements for the three types of surfaces and the influence of flatten order. The denotations (**a–d**) represented four different positions randomly selected on these surfaces, respectively.

|  | Flatten Order | (a) | (b) | (c) | (d) | Average |
|---|---|---|---|---|---|---|
| Artificial surfaces | 0 | 4.73 | 2.49 | 4.27 | 4.45 | 3.99 |
|  | 1 | 6.97 | 6.93 | 8.83 | 10.47 | 8.30 |
|  | 2 | 7.94 | 13.13 | 12.68 | 14.14 | 11.97 |
| Silver thin film | 0 | 3.54 | 3.95 | 3.46 | 3.46 | 3.60 |
|  | 1 | 6.01 | 7.10 | 6.61 | 3.88 | 5.90 |
|  | 2 | 9.34 | 11.74 | 12.27 | 10.56 | 10.98 |
| Milled workpiece | 0 | 8.13 | 6.92 | 3.60 | 4.66 | 5.83 |
|  | 1 | 22.17 | 17.01 | 11.30 | 25.30 | 18.94 |
|  | 2 | 23.51 | 17.39 | 13.70 | 29.04 | 20.91 |

Table 1 shows that the minimum sampling pixels increased along with the increasing flatten order for all the samples, which was consistent with the curve trends shown in Figures 6–8. All the minimum sampling pixel values were low. For all the flatten orders of 0, 1 and 2, the average value was generally below 20. Particularly, the minimum sampling pixels of artificial fractal surfaces and silver thin film surfaces could be even lower than the average value (approximately below 10). Therefore, the $Rq$ value could be regarded as a robust variable against low pixels for a surface. When the flatten order was raised, the $Rq$ values were decreased, which was consistent with the result of our study on thin film surfaces [19]. The decreasing $Rq$ could be attributed to the removal of features when the flatten order was increased. However, the $Rq$ values obtained using different flatten orders could be useful in practical studies, e.g., order 0 was used in the scaling study of roughness by Miyata et al. [49]. Thus, although the pixels of the sub-images of the RSE method would be lower than those of the images measured by the TR method, the $Rq$-$L$ relationship could be reliable due to the robustness of the $Rq$ value against low number of picture elements.

*3.2. Comparison of RSE and TR methods*

To further verify the accuracy of the RSE method and the influence of flatten order, a simulation of surface measurements was conducted by using the artificial fractal surfaces with high pixels (8192 × 8192), whose $L$ was set as 80 μm.

The images with pixels of 512 × 512, which is typically used in AFM measurements, were extracted from high—resolution images. The $L$ values of the extracted were 80, 40, 20, 10 and 5 μm, respectively. Image extraction, i.e., simulated measurement, was repeated five times at each $L$. Then, fractal analysis was carried out on these extracted images by the TR and RSE methods with various flatten orders.

When sub-images were not flattened in the RSE method (RSE-f0), $Rq$-$L$ curves were directly calculated for the extracted sub-images. The typical curves of an artificial surface with $D_i$ = 2.4 were plotted in Figure 9.

In the double logarithmic coordinates, $Rq$-$L$ curves were linear only in the left region, where $L$ was smaller, and was similar to our previous study [34] and that of Miyata et al. [49]. Thus, fractal analysis was carried out for such regions. The linear regions of the five curves could overlap, indicating a similar slope, i.e., a similar calculated fractal dimension ($D_c$). Moreover, the linear regions of the RSE-f0 curves were generally parallel to the $Rq$-$L$ curve of the TR method, which was denoted by the dash line in Figure 9.

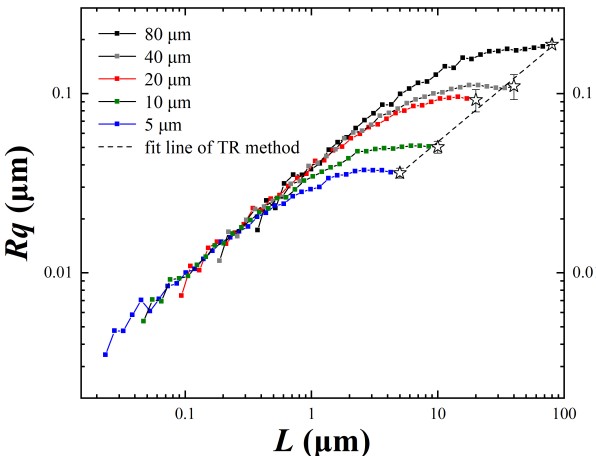

**Figure 9.** *Rq-L* curves calculated with RSE-f0 method for the morphological images (512 × 512) with various *L*, which were extracted from an artificial fractal surface (8192 × 8192) with $D_i$ = 2.4.

Figure 10 shows that when the sub-images were flattened with order 1 or 2 in the RSE method (RSE-f1 or RSE-f2), the *Rq-L* curve of the RSE method could overlap with that of the TR method, in which the same flatten order was utilised for all the as-measured images. The *Rq-L* curves of RSE-f1 and RSE-f2 were generally linear within a broad *L* region of two orders of magnitude.

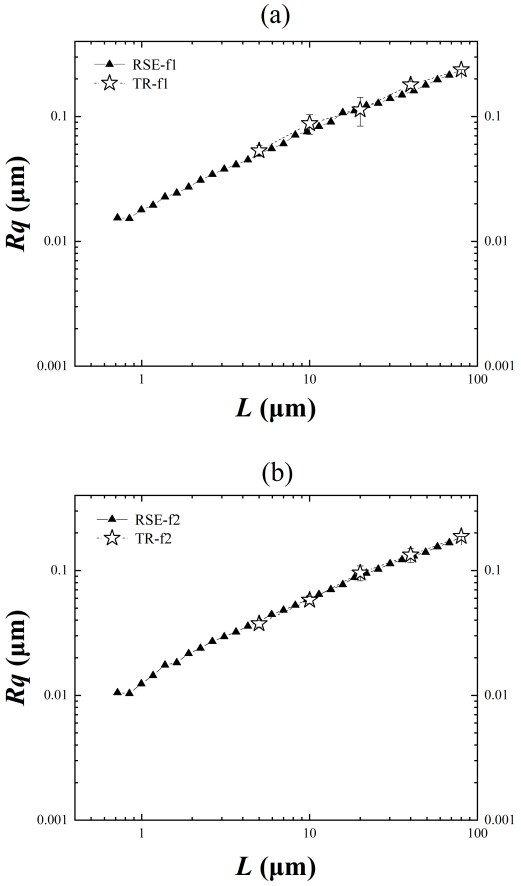

**Figure 10.** (**a**) *Rq-L* curves calculated with RSE-f1 and TR methods; (**b**) *Rq-L* curves calculated with RSE-f2 and TR methods.

The RSE-f0, RSE-f1, RSE-f2 and TR methods were compared, and the $D_c$ values obtained by these methods are summarised in Table 2. The quantified variable and MRE were calculated by using Equation (5) and are illustrated in Figure 11.

$$\text{MRE} = \frac{|D_c - D_i|}{D_i} \times 100\% \tag{7}$$

**Table 2.** $D_c$ values obtained with RSE-f0, RSE-f1, RSE-f2 and TR methods for the images extracted from artificial fractal surfaces with $D_i$ = 2.2, 2.4, 2.6, 2.8.

| $D_c$ | $D_i = 2.2$ | $D_i = 2.4$ | $D_i = 2.6$ | $D_i = 2.8$ |
|---|---|---|---|---|
| RSE-f0 | $2.2083 \pm 0.0155$ | $2.4020 \pm 0.0093$ | $2.6051 \pm 0.0050$ | $2.7872 \pm 0.0020$ |
| RSE-f1 | $2.1928 \pm 0.0070$ | $2.3979 \pm 0.0041$ | $2.5984 \pm 0.0029$ | $2.7888 \pm 0.0018$ |
| RSE-f2 | $2.2036 \pm 0.0034$ | $2.3948 \pm 0.0030$ | $2.5998 \pm 0.0032$ | $2.7853 \pm 0.0022$ |
| TR | $2.1733 \pm 0.0081$ | $2.4283 \pm 0.0107$ | $2.5496 \pm 0.0173$ | $2.8086 \pm 0.0022$ |

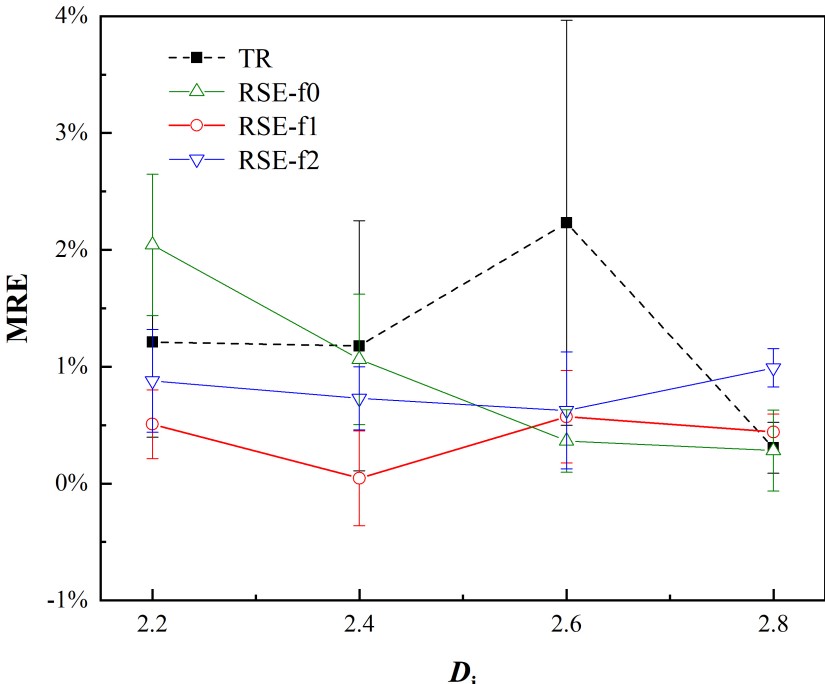

**Figure 11.** The MRE values for the comparison of $D_i$ and $D_c$ obtained with different methods, which were listed in Table 2.

As shown in Figure 11, the RSE-f1 method is the optimal method because its MRE values are below 0.5%, whereas those of the TR method were above 1% in most cases.

## 4. Discussion

The purpose of robustness analysis for surface roughness in this study was to interpret the feasibility of RSE method. Figures 6–8 show that the $Rq$ value of a surface could be robust within a large range of low pixels. Thus the $Rq$-$L$ curves of the RSE method could be similar to those of the TR method, as shown in Figure 9. This similarity of the $Rq$-$L$ curve obtained by the RSE and TR methods could account for the validation of the RSE method. Moreover, because the operation of the RSE method was based on a single morphological image [34], it could be faster than the TR method,

in which over 10 images (25 images in this study) were required to obtain the *Rq-L* curve. Therefore, the RSE method could be regarded as superior to the TR method because of its advantages in both accuracy and efficiency.

The error bar of the *Rq* value for a surface is always large, as shown in Figures 6–9. This significant fluctuation could be regarded as an intrinsic property of surface roughness because it exists in various surface types, which may be attributed to randomness. Therefore, multiple measurements of *Rq* should be conducted to obtain its average value and alleviate the error. In the TR method, multiple actual measurements for each *L* were carried out, thereby increasing the workload of fractal analysis. In the RSE method, multiple measurements could be accomplished by segmenting more sub-images at random positions (100 sub-images at each *L* in this study). The reliability of the RSE method could be based on both the robustness of *Rq* and the extraction of multiple sub-images at each *L*.

The robustness analysis of *Rq* in Section 3.1 was not based on the assumption that a surface is fractal in nature within the entire research scope. For the surfaces of the silver thin film and milled workpiece, the power-law relationship shown in Equation (2) could be applied merely within a certain *L* range, which would be further studied and reported in our future article. The focused aim of this study is to verify the feasibility of RSE method in obtaining the *Rq-L* relationship by a direct comparison against TR method. To support the results obtained from the previous data of artificial surfaces, we further added the comparison based on the AFM data of the silver thin film.

As shown in Figures 12 and 13, the AFM images with *L* of 1, 3 and 10 μm are utilized to calculate *Rq* values, in which 5 positions were randomly selected at each *L*. Then the *Rq-L* curves were calculated with RSE-f0/f1/f2 method. It could be observed that the *Rq-L* curves of RSE-f0 method would deviate from actual AFM measurements, which was consistent with the results of artificial surfaces shown in Figure 9.

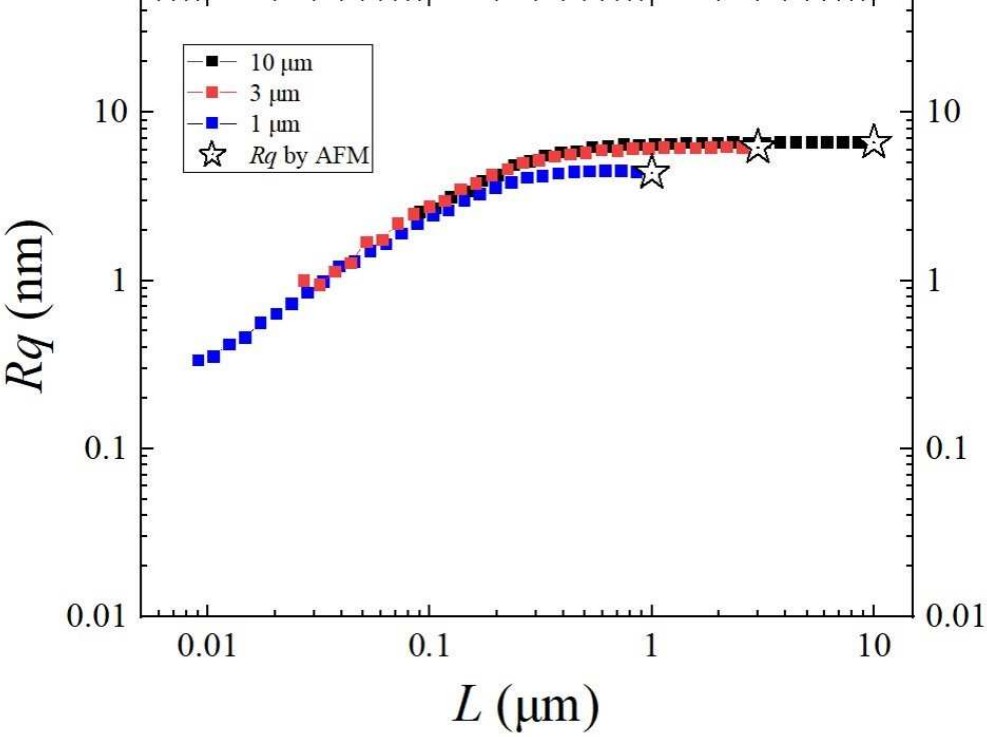

**Figure 12.** *Rq* values obtained by actual AFM measurements at *L* of 1, 3 and 10 μm (denoted with star symbols) and *Rq-L* curves obtained with RSE-f0 method (denoted with squared symbols).

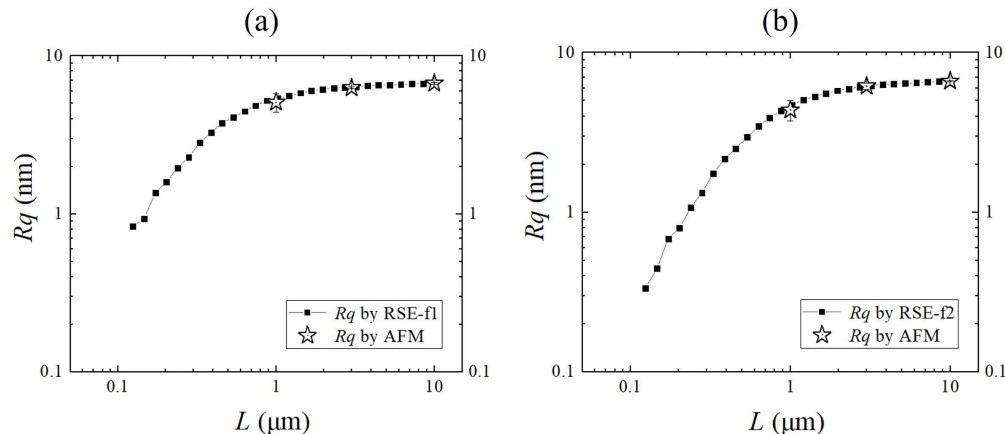

**Figure 13.** *Rq* values obtained by actual AFM measurements at *L* of 1, 3 and 10 μm (denoted with star symbols) and *Rq-L* curves obtained with (**a**) RSE-f1 and (**b**) RSE-f2 methods (denoted with squared symbols).

As shown in Figure 13, the *Rq-L* curves of RSE-f1 and RSE-f2 method could overlap the actual AFM measurements, which were also consistent with the results of artificial surfaces shown in Figure 10. The above results suggested that the flatten modification is necessary in RSE method to obtain the *Rq-L* curves accurately and rapidly.

Therefore, the RSE method is not limited in fractal analysis, but could help collect reliable *Rq* values at small *L* out of a morphological image with a large *L*. this approach helped in our recent study [50] on scaling analysis on the electro-polishing technique.

Finally, the robustness of *Rq* could be preserved if the surface images were randomly sampled. In this study, the sampling process employed the equal interval, which means that each interval between two adjacent data points along *x/y* axis was equal. In our future studies, the influence of random sampling would be analysed to further optimise the operation procedure of the RSE method.

## 5. Conclusions

In this study, the variation of *Rq* value along with the lowering of the number of picture elements was investigated, where the robustness of *Rq* value was found to interpret the validation of the RSE method. Then, a direct comparison of RSE and TR method was performed, where the advantages of both accuracy and efficiency of RSE method could be verified by using artificial surfaces and real surfaces. The above two research aspects could help to demonstrate the credibility and application of the RSE method, which could be superior relative to the traditional methods within the research scope of this study. The conclusions obtained are as follows:

1. For artificial fractal surfaces, that is, the surfaces of a silver thin film and a milled workpiece, reliable *Rq* values could be obtained using the images with low pixels, such as $20 \times 20$ (refer to the maximum average in Table 1).
2. The minimum sampling number of picture elements of RSE-f0 (about 6) was the lower than that of RSE-f1 and RSE-f2 (about 20 and 21, respectively), indicating that it had the best robustness. However, the deviation of the *Rq-L* curve at a large *L* range could cost the accuracy of the RSE-f0 method.
3. A direct comparison between the RSE method and the TR method was carried out. As indicated by the MRE values, the RSE method could be regarded as superior because it is more accurate and efficient than the TR method, especially the error of RSE-f1 and RSE-f2 of the whole fractal dimension calculation are both less than 1.5%.
4. The investigation of the robustness of *Rq* could help reconsidering the scaling property of a surface, regardless of its fractal nature.

**Author Contributions:** W.Z., F.F. and X.L. carried out the research and wrote the original manuscript. X.Q., T.Q. and P.F. assisted with the investigation conceiving. J.H. and H.Z. assisted with the data analysis and manuscript editing. All authors have read and agree to the published version of the manuscript.

**Funding:** This study was supported by National Natural Science Foundation of China under Grant No. 61905129, Guangdong Basic and Applied Basic Research Foundation under Grant No. 2020A1515011199, and Shenzhen Foundational Research Project (Discipline Layout) under Grant No. JCYJ20180508152128308.

**Acknowledgments:** The authors thank Xiangsong Zhang, Binbin Liu and Shaocong Wang of Tsinghua University for their efforts to establish the roughness scaling extraction algorithm.

**Conflicts of Interest:** The authors declare that they have no affiliations with or involvement with organizations that have financial interests in the subject matter or materials discussed in this paper.

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
