# Peer review of "Robustness of Surface Roughness against Low Number of Picture Elements and Its Benefit for Scaling Analysis"

_coatings, doi:10.3390/coatings10080776_

Round 1
Reviewer 1 Report
This paper describes relationships between size of 3D surface height data and the root mean square height calculated from the data, and the manuscript is well written. However, I regret that the work does not include any new approaches and new findings both in terms of mathematics and in practical engineering, and therefore I do not recommend this work for publication in Coatings. Some critical comments and questions follow.
- What is the originality of the work? Which new findings does this work provides? The behaviors of processed data that the authors present in this paper can well be predicted from the existing knowledge of stochastic analysis and the concept of fractal - by using, say spectral density function, auto-correlation function and structure function. It is natural that Rq is not stable for smaller samples and it depends on the correlation length which depends of surface.
- The authors should clearly explain an engineering purpose of discussing the root-mean-square height and the engineering benefit or future benefit for doing what they call the scaling extraction method?
- What do the authors mean by "accuracy" of the method? Please define.
- The authors discuss both the range of stochastic self-affine fractals and the (range of) robustness. Please explain them individually or their relationship.
- How do the authors can ensure the ergodicity of the samples analyzed?
- How many sub images (for each of sub image size) did the authors analyze in the data shown in figures after Figure 6?
- Please give details about the method of flattening and the definition of the flattening order. More importantly, please explain the mathematical effect of flattening?
Reviewer 2 Report
In this paper, a rare method called roughness-scaling extraction (RSE) was used to calculate fractal dimension of roughness of artificial surfaces and the surfaces of two actual samples: silver thin film and aluminum alloy workpiece. This method was compared with traditional method (TR) based on dependence of surface roughness of the scale factor. It was shown that in terms of the accuracy of calculations, the RSE-method is comparable with the TR-method. Moreover, the proposed method is faster because it requires less measurement.
However, when considering the article, unclear questions arose, as follows:
Fig. 1. Comment: The units of x, y and z (height) should be indicated in the caption or in the text.
Lines 90-91: When Di was large, the surface morphology became increasingly irregular and fragmented. Comment: It seems that this statement is not supported by the results of Fig. 1. Visually, the opposite pattern is observed, namely if Di=2.2 (Fig. 1, a), then z-function is irregular, fragmented and has high deviation. While if Di=2.8 (Fig. 1, d) then z-function is smoother and has lower deviation.
Fig. 5, etc. Comment: Why are there no points on the Rq-L graph at L< 1000 nm, when the TR-method was used? To justify the proposed RSE-method, it is necessary to calculate additional points by the TR-method at L< 1000 nm.
The decision: this article is interesting and can be published after minor revision.
Reviewer 3 Report
This paper presents the analysis of the robustness of surface roughness parameter Rq against a low number of picture elements. The Authors compared the calculation accuracy of Rq-L curve between roughness scaling extraction method and traditional roughness method. The research topic has high implications to practical applications and is well suited in the journal scope.
The manuscript consists the results of analytical analyses. The results obtained are well interpreted. The design of the analyses is reasonable and the conclusions are supported by the data.
Structure of this manuscript and quality of graphics are on acceptable level.
I feel that this article can be accepted for publication.
Some of minor comments which should be solved before publication are listed below:
line 20: "Zhang [8]" must be corrected as "Zhang et al. [8]". Check the whole article to avoid similar mistakes.
line 52: "There is a mistake in a surname of Ponomareva.
line 87: Φn should be corrected as Φmn.
equation (1): parameters γ, m and n have not been explained in the text.
Figures 7 and 8: Figures (a) - (d) must be revealed in figure caption.
The header of Table 1 should be corrected. It is not clear what (a)-(d) refer to.
Why Rq parameters were selected for analysis? According to the figures 1 and 4 determination of Sa or Sq parameters seems to be more accurate.
It is suggested to add quantitative conclusions in section 5.
